

# Multi-schema computational prediction of the comprehensive SARS-CoV-2 vs. human interactome

Kevin Dick[1,2], Anand Chopra[3,4], Kyle K. Biggar[3,4] and James R. Green[1,2]

[1] Department of Systems and Computer Engineering, Carleton University, Ottawa, Ontario, Canada
[2] Institute for Data Science, Carleton University, Ottawa, Ontario, Canada
[3] Institute of Biochemistry, Carleton University, Ottawa, Ontario, Canada
[4] Department of Biology, Carleton University, Ottawa, Ontario, Canada

Corresponding author
James R. Green,
jrgreen@sce.carleton.ca

## ABSTRACT

**Background.** Understanding the disease pathogenesis of the novel coronavirus, denoted SARS-CoV-2, is critical to the development of anti-SARS-CoV-2 therapeutics. The global propagation of the viral disease, denoted COVID-19 ("coronavirus disease 2019"), has unified the scientific community in searching for possible inhibitory small molecules or polypeptides. A holistic understanding of the SARS-CoV-2 vs. human inter-species interactome promises to identify putative protein-protein interactions (PPI) that may be considered targets for the development of inhibitory therapeutics.

**Methods.** We leverage two state-of-the-art, sequence-based PPI predictors (PIPE4 & SPRINT) capable of generating the comprehensive SARS-CoV-2 vs. human interactome, comprising approximately 285,000 pairwise predictions. Three prediction schemas (*all*, *proximal*, *RP-PPI*) are leveraged to obtain our highest-confidence subset of PPIs and human proteins predicted to interact with each of the 14 SARS-CoV-2 proteins considered in this study. Notably, the use of the Reciprocal Perspective (RP) framework demonstrates improved predictive performance in multiple cross-validation experiments.

**Results.** The *all* schema identified 279 high-confidence putative interactions involving 225 human proteins, the *proximal* schema identified 129 high-confidence putative interactions involving 126 human proteins, and the *RP-PPI* schema identified 539 high-confidence putative interactions involving 494 human proteins. The intersection of the three sets of predictions comprise the seven highest-confidence PPIs. Notably, the Spike-ACE2 interaction was the highest ranked for both the PIPE4 and SPRINT predictors with the *all* and *proximal* schemas, corroborating existing evidence for this PPI. Several other predicted PPIs are biologically relevant within the context of the original SARS-CoV virus. Furthermore, the PIPE-Sites algorithm was used to identify the putative subsequence that might mediate each interaction and thereby inform the design of inhibitory polypeptides intended to disrupt the corresponding host-pathogen interactions.

**Conclusion.** We publicly released the comprehensive sets of PPI predictions and their corresponding PIPE-Sites landscapes in the following DataVerse repository: https://www.doi.org/10.5683/SP2/JZ77XA. The information provided represents theoretical modeling only and caution should be exercised in its use. It is intended as a resource for the scientific community at large in furthering our understanding of SARS-CoV-2.

# INTRODUCTION

The novel coronavirus (CoV) pandemic has galvanized the research community into the investigation of the SARS-CoV-2 virus and the COVID-19 disease it manifests in humans (*Guarner, 2020*). Research has progressed with unprecedented speed in large part due to the rapid determination of the SARS-CoV-2 genome and proteome. These data enable the research community to collectively contribute to the study and understanding of SARS-CoV-2 and its disease pathogenesis. Given the emergence of three human coronaviruses (HCoVs) causative of severe disease of epidemic or pandemic proportions within the last two decades, we must expand our fundamental understanding of these viruses to rapidly identify putative therapeutic targets, facilitate complimentary research, and inform public discussions for the present and any future outbreaks of HCoVs.

Coronaviruses share many similarities to the influenza viruses in that they are both enveloped, single-stranded, and helical RNA-viruses among the Group IV viral families (*Baltimore, 1971*). The four coronaviruses known to commonly infect humans are believed to have evolved such that they maximize proliferation within a population. This evolved strategy involves sickening, but not ultimately killing, their hosts. By contrast, the two prior novel coronavirus outbreaks (SARS and MERS) arose in humans after cross-species jumps from animals, as was H5N1 (the avian influenza). These latter diseases were highly fatal to humans, with relatively few mild or asymptomatic cases. A greater proportion of mild or asymptomatic cases would have resulted in wide-spread disease, however, SARS and MERS each ultimately killed fewer than 1,000 people (*World Health Organization, 2020*; *World Health Organization, 2011*).

All known HCoVs arise from zoonotic origins (i.e., from other animal species). The wide diversity of CoVs within the animal kingdom stem from the genetic alterations to CoV genomes through acquisition of mutations and a high frequency of recombination between different CoV genomes (*Makino et al., 1986*; *Van Der Most et al., 1992*). Such genetic modifications occurring in animal CoVs may facilitate a ''host jump'' and are the primary reason for inter-species and animal-to-human transmission (*Cui, Li & Shi, 2019*). The HCoVs that are endemic to the human population are causative agents of more mild disease (e.g., common cold) and there is less urgency to identify the animal reservoirs of these viruses.

CoVs are enveloped viruses with a mostly spherical membrane approximately 120 nm in diameter and comprised of 4–5 structural proteins. The single-stranded RNA genome is encapsulated by the Nucleocapsid (N) protein, which functions to package the viral genome into CoV particles during assembly (*Chang et al., 2006*). The Membrane (M) protein plays a central role in assembly of the viral particles, largely by promoting membrane curvature (*Neuman et al., 2011*). The Envelope (E) protein is multi-functional, playing key roles in viral assembly and maintenance, such as mediating ion-channel activity (*Schoeman*

*& Fielding, 2019*). The large membrane projections are trimers of the Spike (S) glycoprotein, responsible for attachment and entry into target cells. Additional smaller 8 nm projections are inherent to lineage A $\beta$CoVs, due to the presence of hemagglutinin esterase (HE) dimers.

It is of critical importance that the cellular entry mechanism and viral replication pathways of SARS-CoV-2 and the role of accessory proteins be rapidly elucidated to develop anti-viral therapies to mitigate the spread and infectivity of the virus in the present pandemic.

Promisingly, many computational approaches have been rapidly deployed to increase our understanding of SARS-CoV-2, including protein function, three-dimensional (3D) protein structures, and possible target regions for small inhibitory molecules (*Senior et al., 2020*; *Smith & Smith, 2020*). Given that the Spike protein from the original SARS coronavirus, SARS-CoV, is known to interact with the human Angiotensin-Converting Enzyme 2 (ACE2), current efforts are focused to better characterize the SARS-CoV-2 Spike protein and its putative interaction with the ACE2 protein.

Similar efforts are being made to understand the functional and evolutionary characteristics of the SARS-CoV-2 proteome, including the determination of evolutionary conserved functional regions between related viruses to inform the use of anti-viral therapeutics (*Cui et al., 2020*). Given the unique infectivity characteristics of this novel coronavirus, the need for effective anti-viral therapeutics is pressing. The long viral incubation period, during which an individual is simultaneously contagious and asymptomatic, has resulted in rapid global proliferation. Leveraging what is known from the original SARS-CoV outbreak and related viral families (previously introduced) this work contributes predicted protein-protein interaction (PPI) networks to guide researchers and form the basis of testable hypotheses warranting wet-lab confirmation.

We hope to contribute to the scientific effort using the latest version of our sequence-based protein-protein interaction (PPI) predictor, PIPE4 (*Dick et al., 2020*) in combination with another state-of-the-art PPI predictor, denoted Scoring PRotein INTeractions (SPRINT) (*Li & Ilie, 2017*). Additionally, we leverage the Reciprocal Perspective (RP) cascaded classification method to further refine predictions (*Dick & Green, 2018*). We leverage a *multi-schema* methodology in order to identify a high-confidence subset of putative interactors. The three predicted interactomes were leveraged in combination to produce candidate targets for experimental validation and to subsequently guide the development of inhibitory polypeptides. Finally, the PIPE-Sites algorithm was used to predict the sub-sequence regions with a high likelihood of mediating the physical interaction between two given pairs (*Amos-Binks et al., 2011*).

Our sequence-based PPI prediction method (PIPE) was previously used during the 2015 Zika virus outbreak to identify putative human-Zika PPIs with the goal of informing rational drug discovery (*Kazmirchuk et al., 2017*). In the present study, of the ~285,000 host-virus pairs, we leverage three prediction schema and two independent PPI predictors to select a highly conservative set of predicted interactions for each of the 14 SARS-CoV-2 proteins considered in this study resulting in the identification of several putative human protein targets. We have publicly released these predictions and related meta-data

for use by the broader scientific community in the following DataVerse repository: https://www.doi.org/10.5683/SP2/JZ77XA (*Dick, Biggar & Green, 2020*).

## METHODS

The multi-schema methodology leveraged in this work follows from and expands upon a previous study of the Zika virus (*Kazmirchuk et al., 2017*) where we defined two initial prediction schemas from which to train the PPI predictors. First, the *all* schema, contains the maximum available number of known virus-human PPIs regardless of the evolutionary distance between those viruses and the target virus (i.e., SARS-CoV-2). This schema groups all viruses into a "viral" collection to serve as a proxy for SARS-CoV-2. The second schema, denoted *proximal*, is a subset of the *all* schema, where only the PPIs from evolutionarily related viruses are considered. In a third schema, denoted *RP-PPI*, both the *all* and *proximal* datasets are leveraged to apply the Reciprocal Perspective cascaded PPI predictor developed by *Dick & Green (2018)*. Specifically, the *proximal* PPIs are used to train the PIPE4 and SPRINT method generating the comprehensive prediction matrix (CPM) representing all possible pairs between the remaining *all* schema pairs and human. From this CPM, the RP features (as described in *Dick & Green (2018)*) were extracted and used to train a downstream model to generate refined predictions between SARS-CoV-2 and human protein pairs.

In the three schemas, as part of an independent evaluation, we remove the previously known SARS-CoV Spike vs. ACE2 interaction to serve as a positive control among the set of predicted interactions. We retained the other four known interactions between SARS-CoV and human within the PPI training set.

The dataset of experimentally elucidated human-virus PPIs was obtained from the VirusMentha database (*Calderone, Licata & Cesareni, 2015*). These 10,693 known PPIs are used to train the PPI predictors and infer new putative interactions between human proteins and the SARS-CoV-2 proteome. For the *all* schema, the proteomes of the 43 viral families were collected from Uniprot and are summarized in the Supplementary Materials. To generate a complimentary predicted interactome using the *proximal* schema, we tabulate the 689 training PPI and the Group IV viral families over which they are distributed (Table 1). Finally, the human reference proteome (UP000005640) was obtained from Uniprot, retaining only the high-quality "Reviewed" Swiss-Prot proteins.

### The SARS-CoV-2 Proteome

The proteome of SARS-CoV-2 was obtained from the Uniprot pre-release available at SARS-CoV-2 Pre-Release, (*Swiss Institute of Bioinformatics, 2020*), with the disclaimer that these data would become part of a future UniProt release and may be subject to further changes. While other SARS-CoV-2 proteins are reported among other sequence repositories, we restricted our study to these highest-confidence proteins available at the time. The 14 SARS-CoV-2 proteins and their function are tabulated in Table 2. Notably, the Spike glycoprotein (Accession: P0DTC2) is of special interest to this and related work, since its SARS-CoV homolog is known to interact with the human ACE2 protein and is presently the target of a recent mRNA-based vaccine candidate.

**Table 1** Group IV Viral Families and their Number of PPIs used in the *Proximal* Prediction Schema.

| Virus Family | Number of PPIs | Capsid Type | Capsid Symmetry | Nucleic Acid Type | Examples |
|---|---|---|---|---|---|
| *Flaviviridae* | 569 | Enveloped | Icosahedral | Single-Stranded | Hepatitis C virus, Zika virus |
| *Togaviridae* | 56 | Enveloped | Icosahedral | Single-Stranded | Rubella virus, Alphavirus |
| *Arteriviridae* | 56 | Enveloped | Icosahedral | Single-Stranded | Arterivirus |
| *Coronaviridae* | 5 | Enveloped | Helical | Single-Stranded | Coronavirus |
| *Hepeviridae* | 3 | Naked | Icosahedral | Single-Stranded | Hepatitis E virus |
| *Astroviridae* | 0 | Naked | Icosahedral | Single-Stranded | Astrovirus |
| *Calciviridae* | 0 | Naked | Icosahedral | Single-Stranded | Norwalk virus |
| *Picornaviridae* | 0 | Naked | Icosahedral | Single-Stranded | Enterovirus, Hepatovirus |

**Table 2** The 14 Proteins in the SARS-CoV-2 Proteome Considered in this Study.

| Uniprot Acc. | Gene Name | Protein Name | Protein Function |
|---|---|---|---|
| P0DTD1 | R1A_WCPV | Replicase polyprotein 1a (R1a) | Viral transcription/replication |
| P0DTC1 | R1AB_WCPV | Replicase polyprotein 1ab (R1ab) | Viral transcription/replication |
| P0DTC2 | SPIKE_WCPV | Spike glycoprotein (S) | Attachement and entry |
| P0DTC3 | AP3A_WCPV | Protein 3a (ORF3a) | ESCRT-independent budding |
| P0DTC4 | VEMP_WCPV | Envelope small membrane protein (E) | ESCRT-independent budding |
| P0DTC5 | VME1_WCPV | Membrane protein (M) | Virion morphegenesis |
| P0DTC6 | NS6_WCPV | Non-structural protein 6 (ORF6) | Unknown; possibly host-virus modulation |
| P0DTC7 | NS7A_WCPV | Protein 7a (ORF7a) | Unknown; possibly host-virus modulation |
| P0DTD8 | NS7B_WCPV | Protein 7b (ORF7b) | Unknown; possibly host-virus modulation |
| P0DTC8 | NS8_WCPV | Non-structural protein 8 (ORF8) | Unknown; possibly host-virus modulation |
| P0DTC9 | NCAP_WCPV | Nucleoprotein (N) | Viral genome packaging |
| P0DTD3 | Y14_WCPV | Uncharacterized protein 14 (ORF8) | Unknown; possibly host-virus modulation |
| P0DTD2 | ORF9B_WCPV | Protein 9b (ORF9b) | Unknown; possibly host-virus modulation |
| A0A663DJA2 | A0A663DJA2_9BETC | Hypothetical ORF10 protein | Presumably not expressed |

## Computational Protein–Protein Interaction Predictors

The computational prediction of PPIs is a diverse field which encompasses multiple paradigms (e.g., sequence-, structure-, evolution-, and network-based methods). Sequence-based predictors rely solely upon primary sequence data, making them amenable to the investigation of proteome-wide networks. Furthermore, these methods tend to be highly efficient, where individual PPIs can be predicted in a fraction of a second.

### The Protein–Protein Interaction Prediction Engine (PIPE4)

PIPE is a sequence-based method of PPI prediction that operates by examining sequence windows on each of the query proteins. If the pair of sequence windows shares significant similarity with a pair of proteins previously known to interact, then evidence for the putative PPI is increased. A similarity-weighted (SW) scoring function uses normalization to account for frequently occurring sequences, not related to PPIs. Given sufficient evidence, a PPI is predicted. PIPE has previously been validated on numerous species for both intra-species and inter-species PPI prediction tasks (*Schoenrock et al., 2011*; *Pitre et*

*al., 2006*; *Pitre et al., 2012*). Furthermore, the distribution of evidence along the length of each query protein forms a 2D landscape that can indicate the site of interaction (discussed later) (*Amos-Binks et al., 2011*).

The fourth version of the Protein-protein Interaction Prediction Engine (PIPE4) was recently adapted to improve predictive performance for understudied organisms (*Dick et al., 2020*). That is, species for which the proteome is known, but the number of experimentally validated intra-specific PPIs is insufficient to train a model to generate the comprehensive interactome. To circumvent this, the PIPE4 algorithm leverages the known PPIs of evolutionarily similar and well-studied organisms, serving as a *proxy* training set. Using an approach denoted as *cross-species* PPI prediction, the experimentally validated PPIs from the proxy species are used to train the PPI predictor which is then applied to the proteome of the understudied target organism. Due to the limited availability of known SARS-CoV-2 PPIs, we here use the PPIs from a collection of well-studied and evolutionarily similar proxy viruses to generate these cross-species predictions as depicted in Fig. 1.

The PIPE4 algorithm is particularly well-suited to cross- and inter-species PPI prediction schemas, given that the SW-scoring function appropriately normalizes the prevalence of sequence windows within each training and target species proteome (*Dick et al., 2020*).

### Scoring PRotein INTeractions (SPRINT)

The SPRINT predictor is conceptually similar to PIPE; SPRINT aggregates evidence from previously known PPI interactions, depending on window similarity with the query protein pair, to inform its prediction scores (*Li & Ilie, 2017*). SPRINT leverages a *spaced seed* approach for determining protein window sequence similarity, where only specific positions in the two windows must be identical as defined by the bits of the spaced seeds. Furthermore, protein sequences are encoded using five bits per amino acid, enabling the use of highly efficient (SIMD) bitwise operations to rapidly compute protein window similarities and, thereby, score predictions (*Li & Ilie, 2017*). The present version of the SPRINT algorithm is not explicitly designed to handle inter- and cross-species prediction, nor to predict the specific subsequence site of interaction between a given pair of proteins. Nonetheless, it is among the only PPI predictors capable of predicting comprehensive interactomes in a timely manner and was demonstrated to outperform other PPI predictors, including the PIPE2 algorithm (*Li & Ilie, 2017*).

### Determining an Appropriate Per-Protein Decision Threshold

For each of the 14 SARS-CoV-2 proteins, we predicted their interaction with each of the 20,366 human proteins resulting in 285,124 unique predictions, forming what we denote the comprehensive prediction matrix (CPM), using each of the two predictors considered. While each method, through a form of cross-validation, might determinate a highly-conservative *global* decision threshold, we know from our work in *Dick & Green (2018)* that such thresholds are sub-optimal. Furthermore, there are insufficient known PPI exemplars between human and SARS-CoV-2 from which to optimize such a threshold. Consequently, for the first time, we employ an RP-inspired method to adaptively determine *local* decision thresholds on a per-protein basis based on the distribution of prediction scores involving each protein.

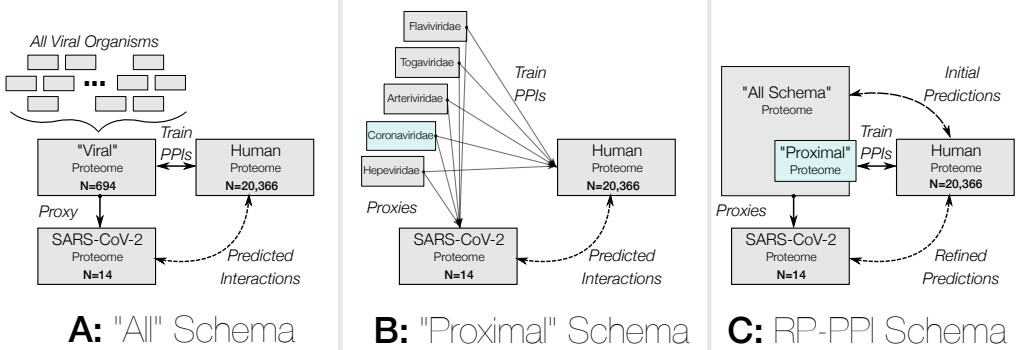

**Figure 1** **Overview of the three prediction strategies to generate the SARS-CoV-2 vs. human interactome.** The three schemas depict how known PPIs are leveraged to train a prediction model to generate predictions for SARS-CoV-2.

From the prediction of all possible pairs, we obtain a CPM. We can then plot the rank-ordered distribution of the putative interaction scores involving each of the *individual* SARS-CoV-2 proteins separately in decreasing rank order by score, forming a *one-to-all* (O2A) score curve. This presents an opportunity to develop protein-specific local decision thresholds, where only those interactions scoring significantly above baseline are reported. These one-to-all score curves are based on the underlying assumption that we expect true SARS-CoV-2 vs. human PPIs to be rare, such that the vast majority of prediction scores should fall below the decision threshold. Furthermore, for the *RP-PPI* schema, we additionally examine the reciprocal perspective, examining one-to-all curves for each human protein and applying analogous decision logic to determine human-protein-specific decision thresholds (*Dick & Green, 2018*).

Thus, for each O2A score curve, a score threshold delineating the "high-scoring" pairs from the baseline was identified and used to determine the high-confidence predicted interactions. In the absence of known PPIs between SARS-CoV-2 and human, it is difficult to determine a suitable global decision threshold. By instead examining the morphology of the O2A score curves for both perspectives, we can qualitatively identify high-scoring pairs. This process can be further automated through the identification of the baseline/knee for each view under the assumption that true PPIs are rare and high-scoring, while non-interacting pairs tend to generate scores residing below the knee in the baseline.

We automated the selection of this operational decision threshold for the 14 SARS-CoV-2 proteins using the Kneedle algorithm (*Satopaa et al., 2011*), applied to its top-1000 predictions, using a sensitivity parameter of 2.0. An example visual illustration of the highly conservative selection of high-confidence interactions is depicted in Fig. 2 and the cut-off scores for each protein are tabulated in the Supplementary Materials.

We identified the common set of predicted pairs above each locally defined knee from *both* the PIPE4 and SPRINT methods (their intersection) for each schema. For example, the *all* schema, resulted in a set of 225 putative human protein targets among 279 intersection pairs. The predicted pairs from each schema were considered to be the

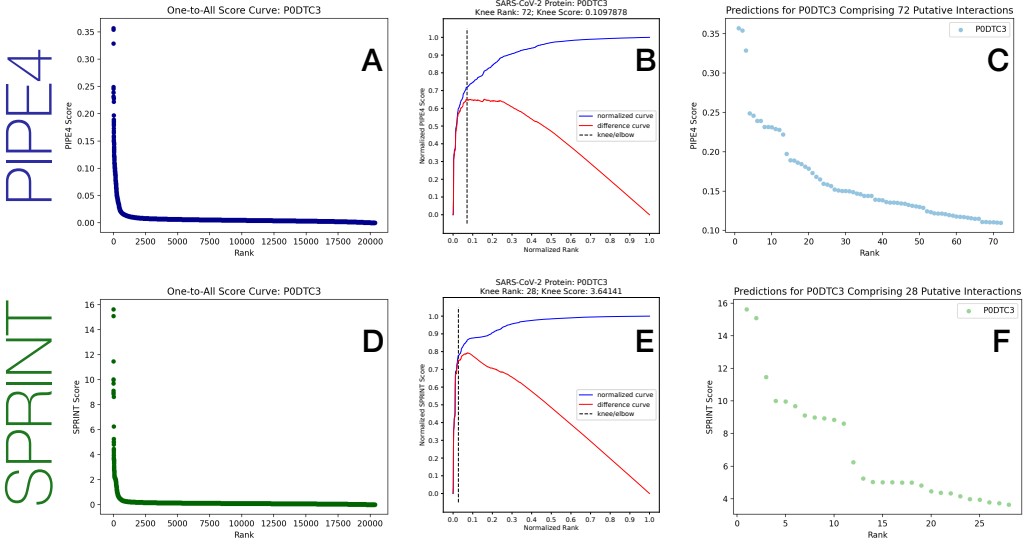

**Figure 2** **Example compilation of the spike protein one-to-all score curve, knee detection for local cut-off, and rank order predictions, for each method.** (A & D) depict the one-to-all score curves from the predicted score between the Spike protein and all proteins in the human proteome. (B & E) depict the detected knee from the top-1000 scores in each one-to-all score curve where the dashed line represents the knee detected using the Kneedle algorithm. (C & F) depict the predicted interactions above the knee.

predicted interactome and were subsequently analyzed by PIPE-Sites; GO-term enrichment analysis was performed using the identified human proteins. The results of each schema's interactome were also combined into higher-confidence sets by taking their set intersections and were visualized as a network.

### Predicting PPI site of interaction using PIPE-Sites & the new similarity weighted landscape

The PIPE4 algorithm generates its prediction for a given pair of proteins based on a two-dimensional landscape of scores, where the score at location $x, y$, the number of sequence window similarity "hits", represents the weight of evidence from the xth and yth subsequence of the human and SARS-CoV-2 proteins, respectively. The PIPE-Sites algorithm examines this landscape and deduces which subsequences from each protein are likely to correspond to the site of interaction (*Amos-Binks et al., 2011*). Such information can guide subsequent detailed investigations to determine the physical binding site which may form the target for novel interventions to disrupt the PPI.

The list of PPIs generated from both methods can be used to inform the design of anti-SARS-CoV-2 therapeutics by using peptide sequences from the predicted PPI site, which we refer to as the PPI-Site. We define the PPI-Site as the peptide sequence that is responsible for mediating a given PPI, which is here estimated using the PIPE-Sites method. A conceptual overview of the PIPE4 landscape matrix and PIPE-Site prediction is illustrated in the Supplementary Materials.

Additionally, we introduce for the first time the Similarity Weighted landscape which is derived from the original PIPE4 landscape with the following modification: the "hits"

representing the weight of evidence from the xth and yth subsequence of the human and SARS-CoV-2 proteins, respectively, are normalized by a cross-species variant of the SW normalization factor in *Dick et al. (2020)* which normalizes window frequency only in species for which there are available PPI training data. This suppresses the effect of highly prevalent windows that are not associated with interactions and amplifies the effect of windows that are relatively rare, yet are frequently occurring in known interactions within the proteomes of species for which training data are available. Specifically, the high-scoring "hot-spots" in the SW landscape are putative subsequences possibly mediating interactions between two proteins. For clarity, we syntactically distinguish the *interface residues* from a predicted *PPI-Site*. Since we are looking at sequence similarity across many proteins, the PPI-Site is a proxy for measuring sequence conservation. Therefore, we are identifying the subsequence that has been conserved to support the interaction site, which may include scaffolding residues distal to the actual interface.

### The reciprocal perspective cascaded classifier: combination of multiple experts

In previous work, we demonstrated that the use of the Reciprocal Perspective PPI cascaded classifier (RP-PPI) produced statistically significant improvement in performance (*Dick & Green, 2018*). Moreover, we here propose the RP-PPI method, as a cascaded machine learning algorithm, can be leveraged to combine features from multiple expert models. Here, for the first time, we jointly combine the features derived from the PIPE and SPRINT models and demonstrate the resulting improvement in performance as part of the *RP-PPI* schema. Furthermore, following from the work of *Kyrollos et al. (2020)*, we implement the cascaded model as an eXtreme Gradient Boosting (XGBoost) regression model (*Chen & Guestrin, 2016*) instead of the Random Forest classifier originally proposed in *Dick & Green (2018)*.

To evaluate the performance increase of the combined classifier, we perform Leave-One-Family-Out cross-validation (LOFOCV), and plot the average Receiver Operating Characteristic (ROC) curve with confidence intervals of one standard deviation. Given certain families had relatively few PPIs, we omitted those with fewer than 50 PPIs from this analysis (a negligible number of pairs were left out). The determination that the combined use of PIPE4 and SPRINT features from their respectively predicted CPMs does, in fact, result in improved performance. We then performed extensive hyper-parameter tuning, evaluated via 10-fold cross-validation, to obtain the most performant model to generate our SARS-CoV-2 vs. human predictions. Varying maximum tree depth ($[3, 4, 5, \ldots, 18]$), number of estimators ($[50, 75, 100, \ldots, 600]$), and the learning rate (9 values considered), we trained and evaluated 29,700 models to arrive to the final model that was used to generate the comprehensive set of prediction as part of the *RP-PPI* schema.

## Gene Ontology (GO) enrichment analysis

To determine which human cellular pathways may be targeted by SARS-CoV-2, PANTHER Gene Ontology (GO) Slim enrichment analysis was applied to each of the predicted interactomes from each schema independently: the 225 human proteins predicted to interact with SARS-COV-2 proteins in the *all* schema, the 123 human proteins in the

*proximal* schema, and the 494 human proteins in the *RP-PPI* schema. The molecular function, biological pathway, and cellular pathway *p*-values were determined with the Fisher's Exact test implemented in the PANTHER GO software (*Mi et al., 2019*). The *p*-values were corrected for multiple testing using the False Discovery Rate (FDR) method described in *Mi et al. (2019)* and significant terms were identified at a threshold of 0.05 and ordered terms by fold enrichment applying variable thresholds.

## RESULTS & DISCUSSION

It is of critical importance that the global research community focus its efforts on the rapid understanding the SARS-CoV-2 virus and the pathogenesis of COVID-19 in order to develop anti-viral therapeutics and additional vaccine targets. Research into *Coronaviridae* biology sharply declined post-SARS-CoV and it is the hope of this work to compliment subsequent primary research in both short-term therapeutic development and long-term COVID-19 symptomology. Fortunately, the prior decades of research into related viral families provide a wealth of data with which to guide current and future studies. For example, the elucidation of the SARS-CoV vs. human inter-species interactome in 2011 using the high-throughput (though false positive-prone) yeast-two hybrid method highlighted cyclophilins as a target for pan-coronavirus inhibitors (*Pfefferle et al., 2011*). Previous knowledge of related coronaviruses within the *Coronaviridae* family provide training samples from which we can identify a number of new high-confidence PPIs that contribute to our understanding of the COVID-19 disease pathogenesis and which may represent targets for novel inhibitory therapeutics.

It is known that the SARS-CoV Spike protein binds to the human ACE2 receptor (*Glowacka et al., 2010*). Upon entry into the respiratory or gastrointestinal tracts, coronaviruses establish themselves by entering and infecting lumenal macrophages and epithelial cells. The viral cell entry program is orchestrated by the spike protein that binds to the human cellular receptors and, thereby, mediates virus-cell membrane fusions.

While the putative interaction between the SARS-CoV-2 Spike protein and human ACE2 receptor is a current focus of the research community, it is also valuable to develop a more holistic understanding of the possibly numerous SARS-CoV-2 vs. human PPIs. Consequently, additional viral-human interactions might be targeted and disrupted with the use of small inhibitory peptides or molecules. To this end, we leveraged sequence-based predictors to score all possible interactions between the SARS-CoV-2 and human proteomes. To identify our highest-confidence set of predictions for the SARS-CoV-2 vs. human interactome, we prepared three training, prediction, and evaluation schemas and combined their predictions to produce a set of candidate interactions for wet-lab validation and the potential design of inhibitory peptides.

### Predictions from the *All* and *Proximal* schemas

As part of the first two schemas (*all* and *proximal*), for each of the 14 viral proteins, we sorted the 20,366 scores (for each human protein) into a monotonically decreasing rank-order which enabled the identification of the subset of high-scoring putative interactors with each viral protein. An example from the *all* schema is depicted in Figs. 2A, 2D.

Rather than apply a globally defined decision threshold (i.e., top-$k$ or minimum threshold), we automatically detected a highly conservative "knee" for each curve (the point of greatest rate of change parameterized by a sensitivity value) to delineate those rare high-scoring pairs from the remaining baseline (Figs. 2B, 2E). For example, within the *all* schema, the union of the $n = 1,209$ predicted PIPE4 and SPRINT high-confidence putative PPIs comprises only $\sim 0.42\%$ of all possible pairs, and their intersection of $n = 279$ putative pairs comprises a highly conservative subset $< 0.098\%$. These data are tabulated in the Supplementary Table, plotted in Fig. 2, and illustrated in Fig. 3A. Taking the combinatorial intersection of the high-confidence predictions from each schema resulted in the highest confidence set of predictions with $n = 7$ predicted pairs (Fig. 3).

## Predictions from the *RP-PPI* Schema

Following from the experimental design of the *all* and *proximal* schemas, the independent predictions from the RP-PIPE4 model and the RP-SPRINT models would have been combined into a single intersection set. However, for the first time, we jointly combined the RP features derived from the PIPE4 O2As with those derived from the SPRINT O2As to train and evaluate a "combination of multiple experts" RP-PPI model. The joint model (using default hyperparameter settings) demonstrated an improvement over the RP-predictor model alone. Interestingly, as illustrated in Fig. S3 the improvement does not appear to be symmetric: the improvement of performance when SPRINT features are joined with the PIPE4 features (Fig. S3: A, blue & grey) is greater than when the PIPE4 features are joined with SPRINT features (Fig. S3: B, blue & grey).

Having established that the combination of experts RP-PPI approach produces improved models, we performed extensive hyperparameter tuning to determine model parameters (550 estimators, maximum tree depth of 17, learning rate of 0.1). Each experiment was evaluated via 10-fold cross-validation with performance measured using the F1 score. Following the training and evaluation of 29,700 models, we identified the best performing model parameters as having a learning rate of 0.1, a maximum tree-depth of 17, and 550 estimators (Fig. S4).

To better understand the features focused upon by the RP-PPI model, we plot the relative feature importance, measured by average information gain in Fig. S5.

Many of the original features from the work of *Dick & Green (2018)* are leveraged in addition to new "statistics-type" features where a given pairs' score is measured in standard deviations away from the identified baseline of a given one-to-all score curve. Notably, baseline scores and ranks for Element A (the SARS-CoV-2 protein) of both methods are among the most distinguishing features (top-4).

With the RP-PPI model, the comprehensive set of human–SARS-CoV-2 pairs were scored to produce 14 one-to-all curves. As above, knee-detection was used to identify the highest confidence subset comprising $n = 539$ pairs, as depicted in Fig. 3C.

We provide the hit and SW landscapes and predicted PIPE-Sites for each of the predicted interactions for each SARS-CoV-2 proteins of each schema. We highlight those 279 pairs within the predicted *all* interactome, the 129 pairs within the predicted *proximal* schema, and the 539 pairs within the predicted *RP-PPI* schema. All data are published in the

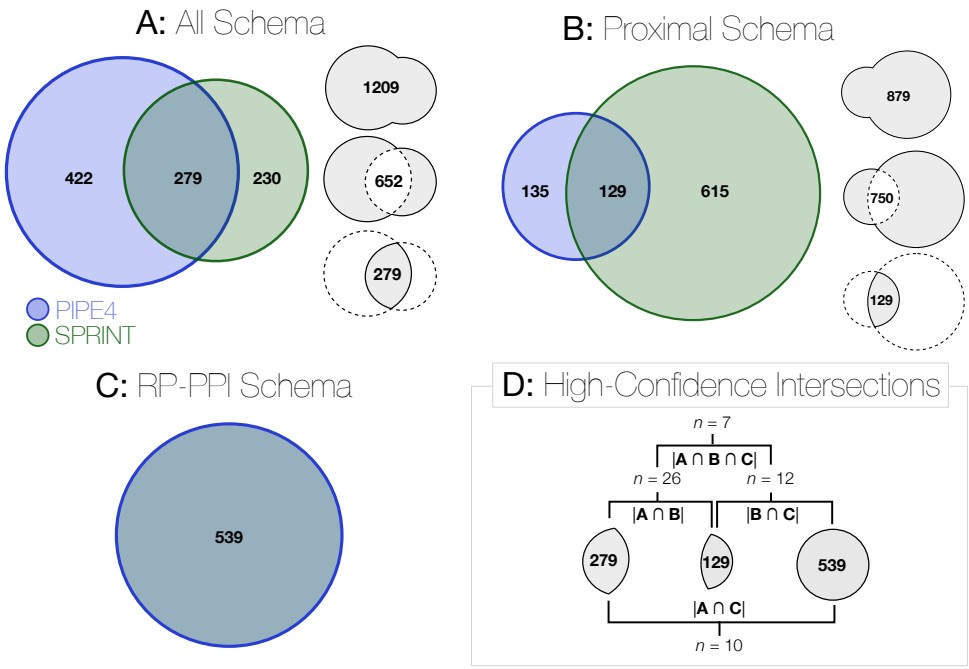

**Figure 3** **Venn Diagram of the human proteins predicted to interact with SARS-CoV-2 proteins.** (A), (B), and (C) depict the number of predicted pairs for each of the schema's putative interactomes. In (D), those interactomes are combined further by taking their intersections with the highest confidence subset comprising $n = 7$ pairs.

following DataVerse repository, for broader use by the scientific community (*Dick, Biggar & Green, 2020*).

We later discuss the biological relevance of our set of highest-confidence predictions and how these may be leveraged to develop anti-SARS-CoV-2 therapeutics. We further consider these interactions in the context of corroborating evidence from scientific literature and illustrate two particular phases of the viral life cylce that might be targeted.

## Putative interaction to target with anti-SARS-CoV-2 therapeutics

The genomes of SARS-CoV-2 and other coronaviruses encode for numerous proteins of diverse functions. The proteolytic cleavage products of the two polyproteins (i.e., non-structural proteins) play essential roles in viral replication but also participate in viral pathogenesis. Similarly, though the structural proteins (e.g., S, E, M, and N) are inherently involved in viral structure and virus-host interactions, such proteins further pathogenesis through interaction with numerous proteins within signaling pathways and, further, accessory proteins are not essential for viral replication; such proteins differ greatly between coronavirus species (*Narayanan, Huang & Makino, 2008*).

To guide the broader research community in expanding the basic understanding of the involvement of SARS-CoV-2 proteins in the underlying pathogenesis of COVID-19, we have visualized the predicted interactomes and incorporated relevant biological information into these networks (Fig. 4). Based on biological process Gene Ontology

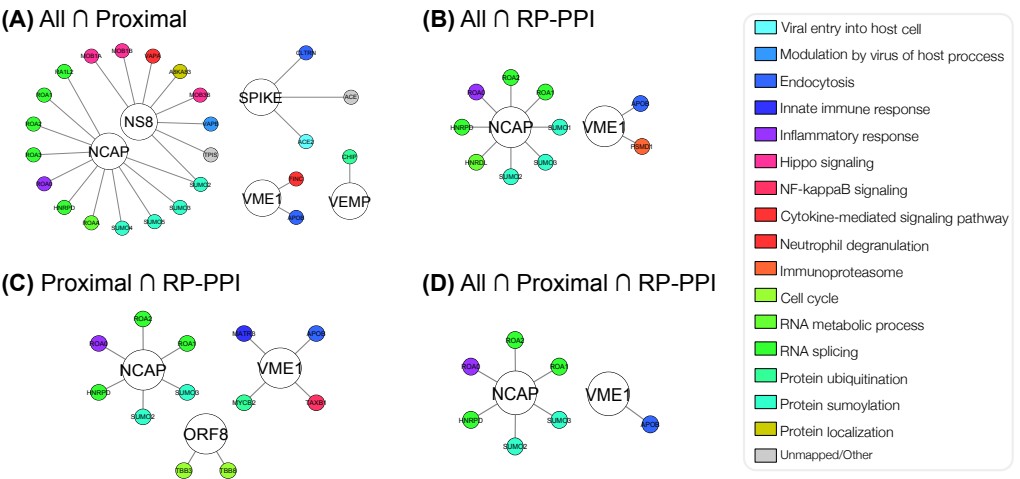

**Figure 4 Network visualization of the highest-confidence predictions.** The colour-function relationship is depicted in Fig. 11. Created using Cytoscape (*Shannon et al., 2003*).

(GO)-terms for each protein within the intersections of the three schemas, individual proteins were manually curated into single descriptors of biological roles, such as those describing viral entry modes, vesicular transport and related processes, types of immune responses, and signaling pathways related to immune responses. Associating proteins with single descriptors is not ideal as numerous proteins possess a broad range of functions. We therefore encourage investigators to assess biological functions of the predicted interactors on an individual basis. In Fig. 5 we illustrate the life cycle of CoVs and highlight the mode of future peptide inhibitors potentially derived from this work.

### The spike protein vs. ACE2 interaction

The PIPE4 and SPRINT predictors scored the SARS-CoV-2 Spike protein vs. human ACE2 protein as the top-ranking prediction in their respective one-to-all score curves (P0DTC2-Q9BYF1) (PIPE4 SW score of 2.159, SPRINT score of 29.3515) within the *all* schema and relatively high-scoring within the other two schemas. As previously noted, this was achieved despite the removal of the known SARS-CoV Spike-ACE2 PPI within the training dataset as part of an independent experiment to determine whether or not the SARS-CoV Spike-ACE2 PPI would have a large effect on scoring this prediction. We further visualize the putative subsequence region of interaction between these proteins in Fig. 6.

Multiple interactions among the high-confidence predictions are biologically relevant when considering known human coronavirus biology. Most notably, predicted within the intersection of the *All* and *Proximal* schemas was the Spike-ACE2 interaction. It is now well-established that SARS-CoV-2 utilizes ACE2 as the main fusion receptor for cell entry (*Hoffmann et al., 2020*). Furthermore, the Spike proteins of SARS-CoV and human coronavirus NL63 interact with ACE2 to facilitate cell entry (*Li et al., 2003*; *Hofmann et al., 2005*). Although the Spike-ACE2 interaction was excluded from our training data, our computational methodology independently predicted the main PPI that permits

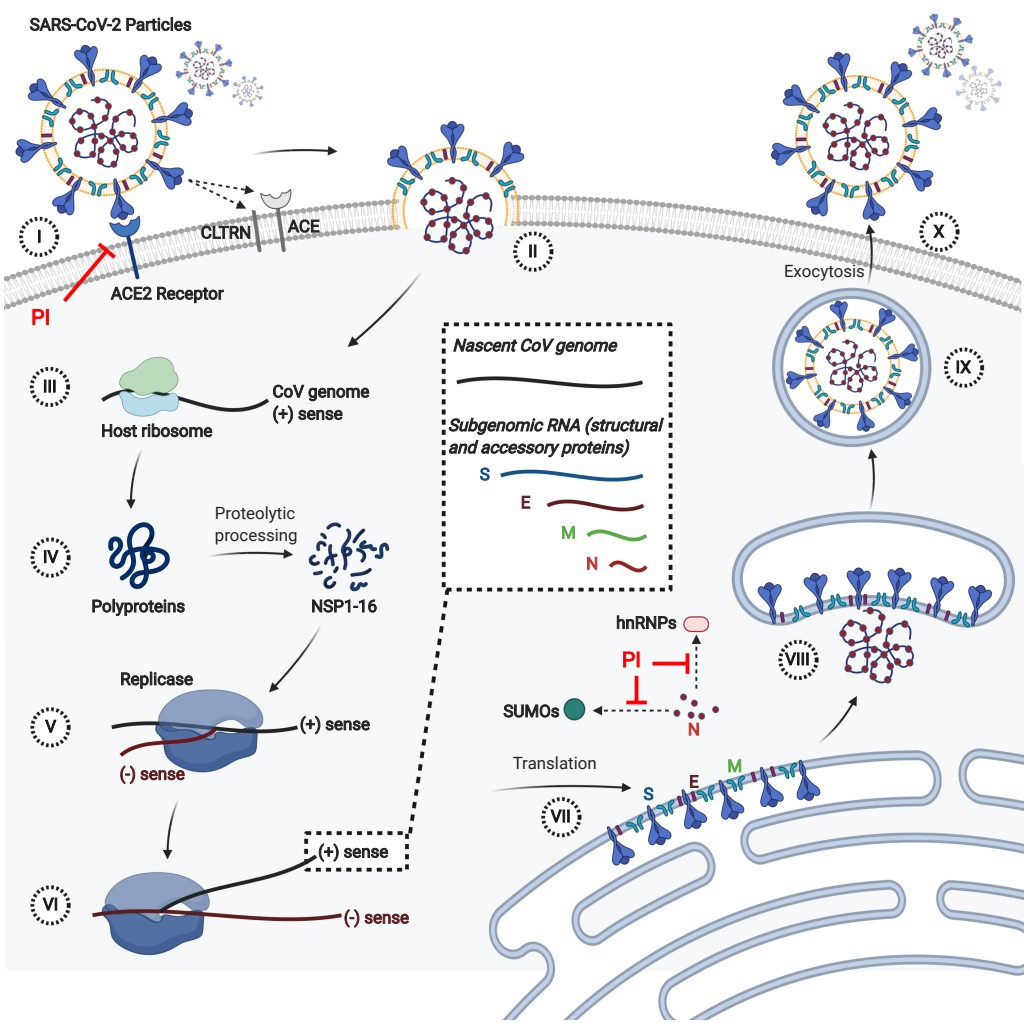

**Figure 5** **Life cycle of CoVs and the mode of future peptide inhibitors (PI) Derived from this Study.** (I) SARS-CoV-2 attaches to the cell surface via interaction of the spike (S) protein with the host ACE2 receptor. (II) The CoV and host membranes coalesce either at the cell surface or within endosomes, releasing the CoV genome into the cytoplasm. (III) Host ribosomes use the CoV genome as a template and translate polyproteins 1a and 1ab. (IV) The polyproteins mature into individual non-structural proteins (NSPs) 1-16 via autoproteolytic processing. (V) Multiple NSPs form a viral replicase complex, which performs negative-strand synthesis of genomeic and subgenomic RNA negative-strand templates. (VI) The viral replicase synthesizes nascent plus-strands of the full-length CoV genome and subgenomic RNAs encoding structural (S, E, M, N) and accessory proteins (not shown). (VII) S, Membrane (M), and Envelope (E) proteins are translated at the endoplasmic reticulum (ER) and inserted into the ER membrane. The Nucleocapsid (N) protein is translated within the cytoplasm. (VIII) The N protein encapsulates the nascent CoV genome and interacts with the other structural proteins within the ER-Golgi intermediate compartment (ERGIC). (IX) Mature CoV particles are formed within vesicles upon budding into the lumen of the ERGIC. (X) CoV particles are released upon exocytosis. Besides the validated S-ACE2 interaction, other notable predicted protein-protein interactions are indicated by dashed arrows. This figure was made in ©BioRender - biorender.com.

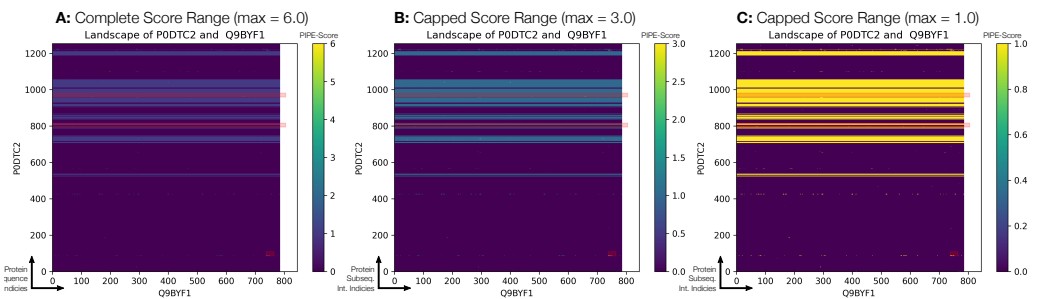

**Figure 6** The PIPE-sites landscape between the SARS-CoV-2 Spike protein and human ACE2 protein. Within each panel, the three red rectangles represent the predicted PIPE-Sites regions. (A) depicts the completely predicted landscape with the complete numerical range of scores depicted. To more easily visualize the high-scoring subsequence regions, (B & C) apply a numerical "capped threshold" where any value greater than or equal to the maximum threshold is set to that value. This has the effect of emphasizing the regions of potential interest. A threshold of 3.0 is applied in B and a threshold of 1.0 in C. See the Supplemental Information for guidance on the interpretation of these landscapes.

SARS-CoV-2 cell entry; therefore this, and similar computational pipelines, hold promise for screening candidate modes of entry of future viruses. Thus, other interactions within these high-confidence intersections of the schemas may be biologically relevant and worthy of further investigation. Besides the Spike-ACE2 interaction, two other high-confidence interactors of Spike were found to be collectrin (CLTRN) and ACE. CLTRN is a homolog of ACE2, lacking the extracellular catalytic domain, sharing 47.8% identity with C-terminal regions of ACE2 (*Zhang et al., 2001*). Furthermore, ACE is homologous to ACE2, sharing 42% identity between catalytic domains (*Donoghue et al., 2000*). This finding corroborates related research reporting that SARS-CoV-2 can infect human respiratory epithelial cells through interaction with the human ACE2 receptor (*Letko, Marzi & Munster, 2020*).

Certainly, if the SARS-CoV-2 and SARS-CoV Spike proteins share sufficient sequence and structural similarity, it can be expected that anti-virals designed against SARS-CoV may also be effective against SARS-CoV-2. We investigate this sequence similarity by performing a BLASTp alignment of the two sequences. Interestingly, only 76% identity was observed (Fig. 7) suggesting that the SARS-CoV-2 spike protein might have evolved to be sufficiently different from its SARS-CoV variant to render existing anti-virals ineffective. Given that the Spike protein is the main point of interface with the host, we can expect that it would be rapidly evolving. The SARS-CoV-2 variant likely shares a similar mechanism of action where the recombinant SARS-CoV-2 spike protein downregulates ACE2 expression and thereby promotes lung injury (*Glowacka et al., 2010*).

Consequently, the elucidation of the Spike-ACE2 binding interface is needed to design novel therapeutics. To that end, we used the PIPE-Sites algorithm to predict the three most likely putative interaction interfaces between the Spike (P0DTC2) and ACE2 (Q9BYF1) proteins (Fig. 6). Note that all predicted subsequence offsets are 0-indexed. With a maximum landscape peak of 6, the PIPE-Sites algorithm identified three putative interaction interfaces:

1. **P0DTC2**: [86–109]; **Q9BYF1**: [738–816]

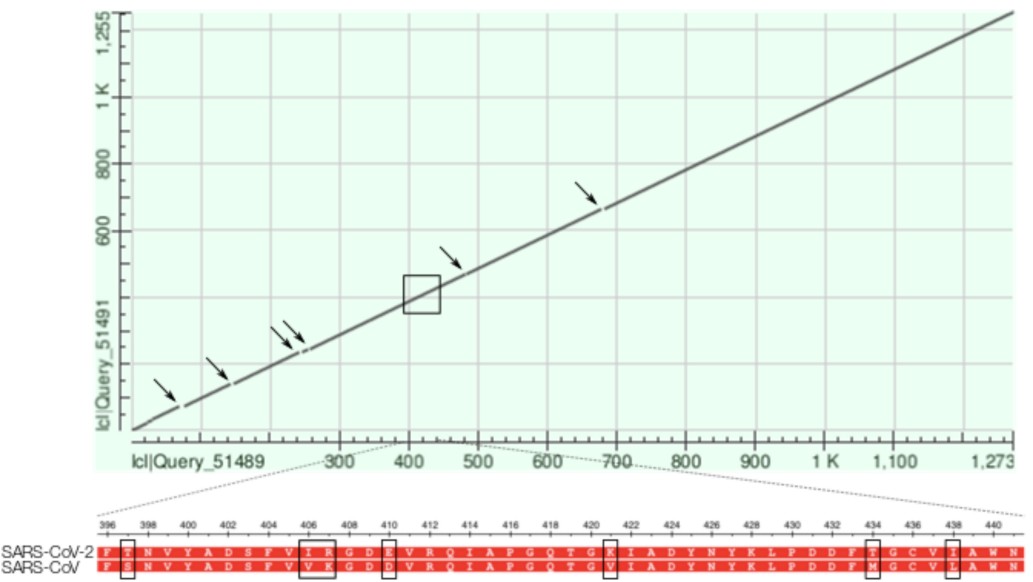

**Figure 7** **Dot plot of the BLASTp alignment of the SARS-CoV and SARS-CoV-2 Spike protein.** The alignment of the two proteins results in a *max score* of 2039, a *total score* of 2039, 100% coverage, an *E-value* of 0.0, and 76.04% identity. Specifically: 971/1277 (76%) identities, 1109/1277 (86%) positives, and 26/1277 (2%) gaps. Arrows indicate gaps within the alignment and the zoomed-in region highlights the six mismatches around residue 420.

2. **P0DTC2**: [795–816]; **Q9BYF1**: Entire sequence
3. **P0DTC2**: [960–981]; **Q9BYF1**: Entire sequence

Interestingly, the PIPE-Sites score landscape in Fig. 6 exhibits a number of horizontal bands indicative of subsequence regions along the Spike protein that correspond to a relatively high likelihood of interaction. While the PIPE-Sites algorithm only identifies three putative regions, these bands suggest additional regions of interest.

The highest-scoring predicted PIPE-Site interface corresponds to the Spike [86–109] subsequence and the ACE2 [738–816] subsequence, which resides within the *intracellular* cytoplasmic domain of ACE2. However, upon closer inspection of other "hot spot" regions within the landscape, we note several that reside within the *extracellular* N-terminal region of ACE2 (i.e., residues ∼[30-84] & [353-357]). In particular, we note the three following regions of interest:

*Visually high-scoring region:* **P0DTC2**: near residue 1224; **Q9BYF1**: [15–23]
*Within ACE2 residues [30-84]:* **P0DTC2**: near residue 420; **Q9BYF1**: [80-84]
*Within ACE2 residues [353-537]:* **P0DTC2**: near residue 420; **Q9BYF1**: [355-357]

Most interestingly, certain of these region along the Spike protein appears to coincide with mismatched or gap regions along the dot plot comparing the SARS-CoV vs. SARS-CoV-2 alignment depicted in Fig. 7). For example, upon closer investigation of the alignment around residue 420, we note six mismatches. Their proximity to a candidate region of interaction certainly warrant additional experimental investigation (Fig. 7).

While numerous inhibitory strategies exist, including the use of small molecules or small interfering RNAs, this research is most directly amenable to the design of small inhibitory peptides that inhibit virus infection by preventing Spike protein-mediated receptor binding and blocking viral fusion and entry (Fig. 5). Unfortunately, much like small peptides and interfering RNAs, peptide-based solutions are disadvantaged by their *low antiviral potency*.

### HLA class I/II histocompatibility antigen

Among the 225 human proteins identified in the *all* schema, six Human Leukocyte Antigen (HLA) class I/II histocompatibility antigens were predicted to interact with P0DTC3, the SARS-CoV-2 Protein 3a (ORF3a):

- **P13747**: HLA-E HLA-6.2 HLAE
- **P01911**: HLA-DRB1
- **P17693**: HLA-G HLA-6.0 HLAG
- **P04439**: HLA-A HLAA
- **P10321**: HLA-C HLAC
- **P30511**: HLA-F HLA-5.4 HLAF

The visualization of the predicted site of interaction for the six HLA interactions highlight a consistent subsequence region of the SARS-CoV-2 protein 3a between amino acids [202–222] (Fig. 8). Literature review reveals that one of the open reading frames (ORFs) of the SARS-CoV virus, the ORF3a, encodes the variant 274 AA-long Protein 3a. A previous study used sequence analysis that suggested that the ORF3a aligned to a calcium pump present in *Plasmodium falciparum* and glutamine synthetase found in *Leptospira interrogans*. This sequence similarity between the three organisms was found to be limited only to amino acid residues [209–264], which form the cytoplasmic domain of ORF3a. This subsequence region was predicted to be involved in calcium binding and then confirmed in vitro (*Minakshi et al., 2014*).

Given the important role that calcium plays as part of virion structure formation, virus entry, viral gene expression, virion maturation, and release, these regions of Protein 3a are of possible interest for disruption of SARS-CoV-2. Specifically, the design of a small inhibitory peptide targeting this subsequence region of Protein 3a might disrupt the viral life cycle.

### Heterogeneous Nuclear Ribonuclear Proteins (hnRNPs)

The Nucleocapsid (abbreviated N or NCAP) protein was predicted to interact with four heterogeneous nuclear ribonuclear proteins (hnRNPs) within the intersection of the three schemas. PPIs between the NCAP protein and those involved in RNA related processes are not surprising, especially considering that NCAP protein of coronaviruses plays a role in viral RNA genome packaging as it is capable of binding single-stranded RNA (*Huang et al., 2004*). Notably, the N protein of SARS-CoV-2 was predicted to interact with hnRNP A1 (i.e., ROA1). This interaction has previously been validated in the context of SARS-CoV NCAP protein and was found to be a high affinity interaction (*Luo et al., 2005*). Additionally, this physical interaction is also inherent to a mouse coronavirus species (i.e., mouse hepatitis virus, MHV) (*Wang & Zhang, 1999*). The role of hnRNP A1 as a host cell factor in MHV

Peer J

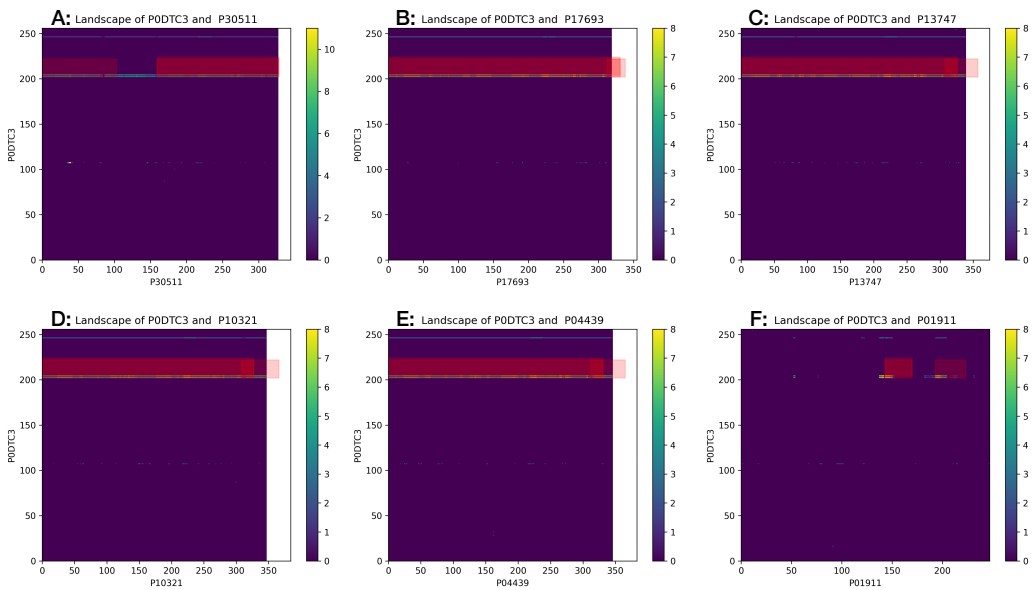

**Figure 8** **Landscapes of the six predicted HLA interactors with SARS-CoV-2 Protein 3a.** The three red rectangles represent the predicted PIPE-Sites regions. They're "shifted" relative to the highlighted cells due to the algorithm's use of a window of 20 amino acids in length that extends both to the left (along the $x$-axis) and upwards (along the $y$-axis). This implementation may also result in the predicted site extending past the coloured matrix, either to the right or above. The PIPE-Sites may overlap when numerous hits appear within close proximity, as is the case when a "band of hits appears in the matrix. See the supplementary material for guidance on the interpretation of these landscapes.

coronavirus biology is not clear, as initially it was shown that this protein functions in MHV RNA synthesis in the cytoplasm, however involvement in these roles (e.g., RNA genome replication and discontinuous transcription) were later contradicted (*Shi et al., 2000*; *Shen & Masters, 2001*). Furthermore, multiple hnRNPs were shown to be upregulated in SARS-CoV infected cells (*Jiang et al., 2005*). The function of hnRNPs in SARS-CoV-2 pathogenesis may relate to previous findings that suggest a role for such proteins in viral RNA synthesis, and therefore these NCAP-hnRNP interactions may present as druggable targets. To that end, Fig. 9 illustrates both the hit and SW landscapes which may serve to identify putative subsequences that may mediate the virus-host interactions.

### Small Ubiquitin-related Modifier (SUMO) Proteins

Lastly, several small ubiquitin-related modifier (SUMO) proteins were predicted interactors within the high confidence intersections. Notably, SUMO proteins were primarily predicted to interact with the NCAP protein. Although interactions are contextually different than post-translational modifications, the SARS-CoV NCAP was previously shown to both interact with an E2 enzyme involved in SUMOylation as well as undergo SUMOylation at the lysine-62 residue (*Fan et al., 2006*; *Li et al., 2005*). SUMOylation of NCAP at lysine-62 promotes homo-oligomerization and this residue may be involved in the disruption of host cell division (*Li et al., 2005*). Whether this SUMOylation occurs within the context of

SARS-CoV-2 and the significance of this remains to be explored; however, based on the previous findings, this interaction should be further investigated.

## GO-Term analysis of human proteins among the predicted interactomes

For each of the schemas, the human proteins within the respective intersections of the PIPE4 and SPRINT predicted interactions were used to run a number of GO-term analyses to better understand the functional role of the human proteins involved. To this end, the GO-Slim Panther Classification System was used to run over/under-representation analysis of the predicted sets of human proteins as compared to the reference human proteome. A Fisher's Exact test with correction for False Discovery Rate was used to extract a list of the most enriched GO-terms among the human proteins for which GO-term data were available. To limit the number of functions, variable thresholds for fold enrichment were applied. For example, among the tables for the *all* schema, the Molecular Functions exhibiting a fold enrichment greater than 3 are reported; the Biological Processes exhibiting a fold enrichment greater than 50 are reported; and the Cellular Components exhibiting a fold enrichment greater than 15 are reported. The fold enrichment cut-offs were selected to limit the size of the tables; the complete tables are available in the appendix of the Supplemental Information and at at public repository, *Dick, Biggar & Green (2020)*.

While this current analysis combines all predicted human interactors together, a more revealing analysis might investigate the resultant GO-terms on a per-viral-protein basis to identify those human pathways and biological processes most sensibly targeted by SARS-CoV-2. This analysis is left to future work.

We encourage the scientific community to delve into the findings of this study. For example, of the GO-terms observed from the *all* schema alone, the highly over-represented biological processes in Supplementary Materials table 4 are interesting. Notably, the top-9 GO-terms have a 96.98 fold enrichment given that the predicted set of human interactors contain all of the proteins from the *H. sapiens* reference (i.e., the number of proteins present in the reference are also in the sample: 2/2, 8/8, and 3/3 among the top-3, respectively). We specifically highlight the "antigen processing and presentation of exogenous peptide antigen via MHC class Ib" (GO:0002477) and the "calcium ion transport from cytosol to endoplasmic reticulum" (GO:1903515). Moreover, the top-ranking cellular component GO-terms (Supplementary Materials tab5) show notable over-representation of "MHC class Ib protein complex" (GO:0032398), "MHC class I protein complex" (GO:0042612), and numerous proteasome complex terms. While only a shallow analysis is presented here, a more involved investigation into these predicted interactions promises to reveal putative targets for novel inhibitory peptides.

## A literature curated subset of candidate human protein targets

In the work of *Gordon et al. (2020)*, $n = 332$ pairs between the $m = 26$ SARS-CoV-2 proteins and $m = 332$ human proteins (i.e., each human protein was involved in exactly one pair) were uncovered. While our highest-confidence interactomes do not predict any of these pairs (i.e., there is, unfortunately, no overlap between the $n = 322$ pairs in the
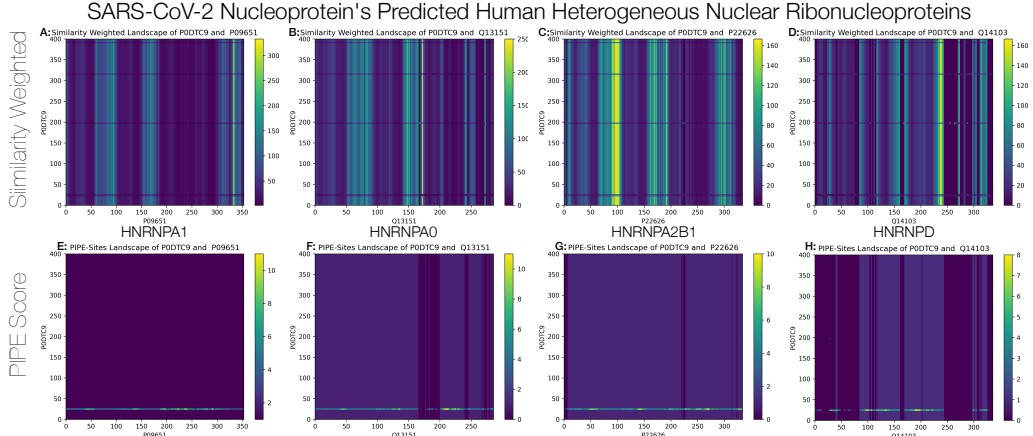

**Figure 9  Hit and SW landscapes for the four predicted hnRNPs to guide the design of peptide inhibitors.** Both "hotspots" and "bands" identify subsequence regions of interest to target with peptide inhibitors. See the Supplemental Information for guidance on the interpretation of these landscapes

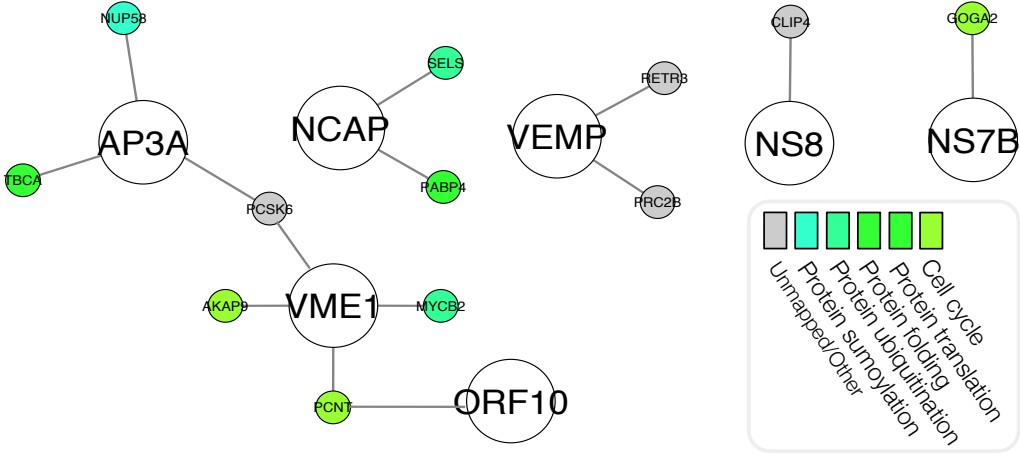

**Figure 10  Network visualization of the 14 predicted pairs involving one of the 332 human proteins from gordon2020sars.** Created using Cytoscape shannon2003cytoscape.

work of *Gordon et al. (2020)* and the $n = 907$ union-of-three-interactome pairs, $A \cup B \cup C$), we emphasize that the work of *Gordon et al. (2020)* considered $m = 26$ proteins of the SARS-CoV-2 proteome while this work comprises only an $m = 14$ subset of those proteins.

As the $m = 332$ human proteins identified in the work of *Gordon et al. (2020)* are of putative interest to the scientific community in an effort to counter the COVID19 pandemic, for convenience, from the union of the three high-confidence interactomes (i.e., $A \cup B \cup C$, notation from Fig. 3D) we extracted any predicted interactions involving one of the $m = 332$ human proteins resulting in the $m = 14$ putative pairs. This set of protein pairs may represent a focused subset of candidate pairs for subsequent investigation. This small network is depicted in Fig. 10.

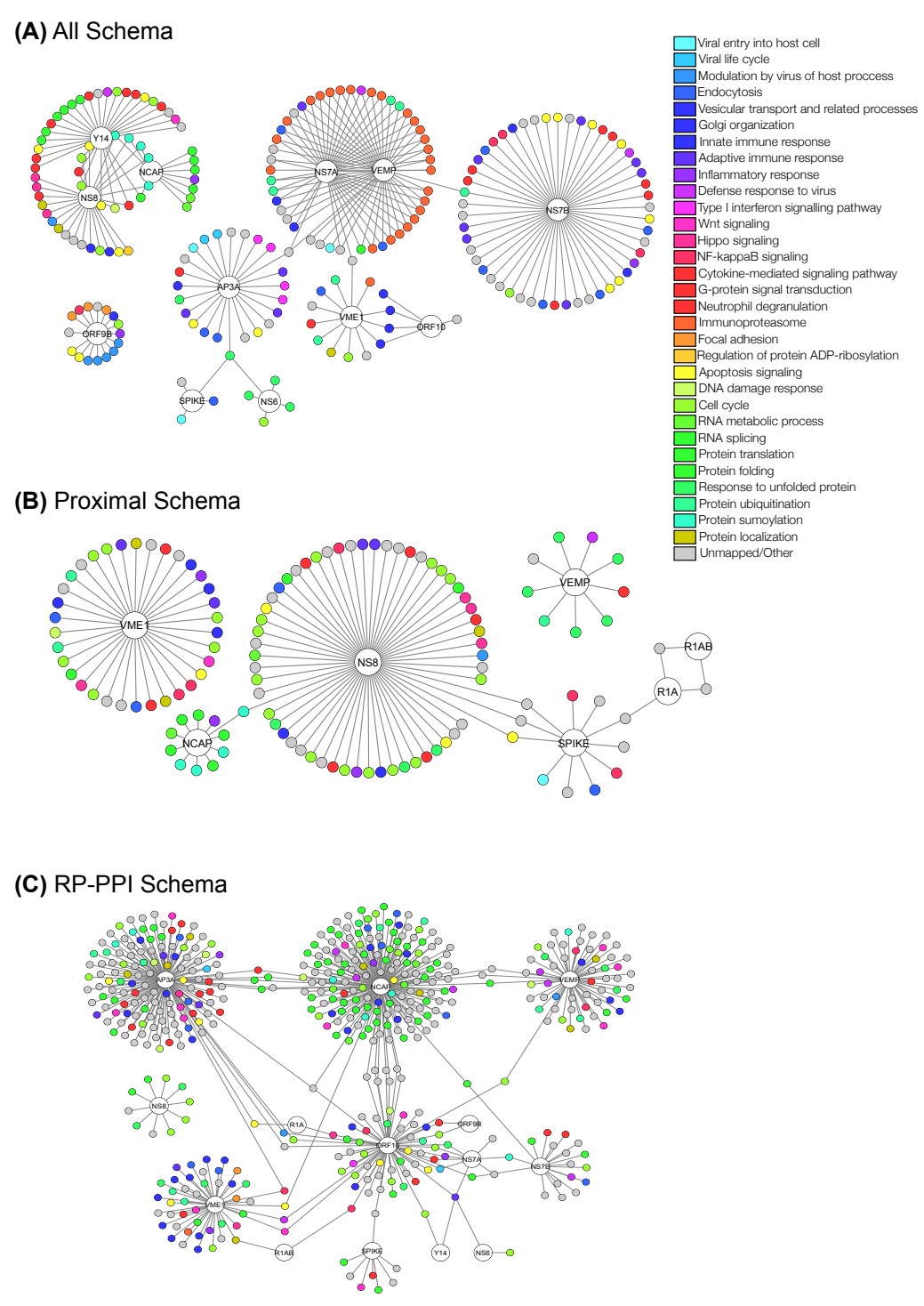

**Figure 11 Network visualization of the complete predicted interactomes for each schema.** (A) All schema. (B) Proximal schema. (C) RP-PPI Schema.

Finally, even if no overlap between Gordon et al. and our predicted interactomes exists, there is, indeed, an overlap of the GO-terms represented by the proteins of both sets. Notably, we identify 220 shared GO-terms between our sets indicative of a large functional overlap including such terms as viral process (GO:0016032), vesicle-mediated transport (GO:0016192), and response to virus (GO:0009615).

## Complete predicted interactomes

To better visualize the predicted interactome the complete network-based representation is depicted in fig:networkfull. Much like the HLA proteins highlighted above, we note a number of highly represented GO-terms around several of the proteins of interest including those related to the immune response, various types of signalling, and the viral life cycle. We hope that this work will guide the broader research community in their search for putative inhibitory molecules.

## CONCLUSIONS

The purpose of this work is to help guide the broader research community in the collective pursuit to understand the SARS-CoV-2 viral pathogenesis. To that end, we assessed 285,124 protein pairs using two state-of-the-art sequence-based PPI predictors within three prediction schemas, thereby creating the comprehensive SARS-CoV-2 vs. human interactome. For each of the 14 SARS-CoV-2 proteins considered in this study, a highly conservative locally defined decision threshold was determined to obtain a predicted interactome comprising putative PPIs within the predicted intersection of the PIPE4 and SPRINT methods. Furthermore, the PIPE-Sites algorithm was used to predict the putative interaction interfaces to identify the subsequence regions of interest that might mediate these interactions.

These predictions have been deposited in a public DataVerse repository for use by the broader scientific community in the collective effort to combat the COVID-19 pandemic (*Dick, Biggar & Green, 2020*). We re-emphasize that the information provided is theoretical modelling only and caution should be exercised in its use. It is intended only as a resource for the scientific community at large in furthering our understanding of SARS-CoV-2.

## ACKNOWLEDGEMENTS

The authors would like to thank Franois Charih for his valuable and constructive suggestions during the planning and development of this research work.

### Funding

This work was supported by a COVID19 Rapid Response Research Grant from Carleton University. There was no additional external funding received for this study. The funders had no role in study design, data collection and analysis, decision to publish, or preparation of the manuscript.

## Grant Disclosures

The following grant information was disclosed by the authors:
Carleton University.

## Competing Interests

The authors declare there are no competing interests.

## Author Contributions

- Kevin Dick conceived and designed the experiments, performed the experiments, analyzed the data, prepared figures and/or tables, authored or reviewed drafts of the paper, and approved the final draft.
- Anand Chopra performed the experiments, analyzed the data, prepared figures and/or tables, authored or reviewed drafts of the paper, and approved the final draft.
- Kyle K. Biggar and James R. Green conceived and designed the experiments, analyzed the data, prepared figures and/or tables, authored or reviewed drafts of the paper, and approved the final draft.

## Data Availability

Protein-protein interaction data are available at the Scholars Portal Dataverse:

Dick, Kevin; Biggar, Kyle K.; Green, James R., 2020, "Comprehensive Prediction of the SARS-CoV-2 vs. Human Interactome using PIPE4, SPRINT, and PIPE-Sites", https://doi.org/10.5683/SP2/JZ77XA, Scholars Portal Dataverse, V1, UNF:6:LaU8vpF6y1UavvDQrlXzpg== [fileUNF].

## Supplemental Information

Supplemental information for this article can be found online at http://dx.doi.org/10.7717/peerj.11117#supplemental-information.

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
