# Peer review of "Multi-schema computational prediction of the comprehensive SARS-CoV-2 vs. human interactome"

_PeerJ, doi:10.7717/peerj.11117_

## Round 0.1 · original submission · Minor Revisions

Thank you again for your submission. There were just a few minor questions brought up by one of the reviewers that, if addressed, may improve clarity of the paper for readers. This a timely and well-written manuscript, so any revisions will not need to be sent out again for review.

Reviewer 1 ·

Basic reporting

The manuscript investigates interactions between the proteins of the SARS-CoV-2 virus and human proteins. Two top protein-protein interaction (PPI) predictors are used and further efforts are made to identify the most confident PPIs. In addition, the prediction are enhanced by the identification of the subsequences that medite the interactions. The main goal is to provide additional information for the community that can be useful in understanding and disrupting its host-pathogen mechanisms.

The manuscript is clearly written, provides appropriate references, includes relevant
figures and makes its data available.

Experimental design

The research described is very well within the scope of the journal. The research is clearly described and important for the problem investigated. The methods are new and interesting, well presented and reproducible.

Validity of the findings

Findings are important, valid and potentially very useful for a large community. To the extent that they overlap, they agree with existing knowledge about the virus.

The overall evaluation of the manuscript is highly positive. It provides an interesting, useful and timely contribution to a subject of high importance.

Reviewer 2 ·

Basic reporting

On page 5, the Kneedle algorithm should be referenced: DOI: 10.1109/ICDCSW.2011.20

In Figure 2, I imagine that the dotted line represents the knee? This should be annotated.

Line 350: "lifecylce" should read: "life cycle"

The color scale of Figure 6 could be adjusted such that differences between the values are more visible. It seems like little to no values are above 3, which could be the maximum of the color scale.

The unit/nature of the values on the legend of Figure 6 and 8 are not reported.

Line 446: there should be a comma before the word "which".

Experimental design

While I am sure that there is a reason for this, why were only 14 proteins of SARS-CoV-2 used for the protein-protein interaction predictions. This should be justified in the manuscript.

This is not a request per se, but I am curious whether the Kneedle algorithm strategy could have been replaced with a null model assessing the statistical significance of a given confidence score. For instance, one could shuffle the protein sequences provided as input and compare the confidence scores of the predicted interactions of those shuffled sequences to the interaction predictions obtained using real sequences. Comparing the number of interactions passing a given confidence score threshold could provide an estimate of the false discoveries. While what I am proposing is a rudimentary approach and that a more sophisticated null model could certainly be designed, I was wondering whether such a general approach would make sense? I would be curious to know what the authors think about this. This could potentially be a discussion point if the authors believe that it is relevant.

Validity of the findings

I understand that there is no overlap with the predictions performed in this manuscript and the Gordon et al., 2020 study of the SARS-CoV-2 and human interactome. However, I was curious to see if there is any overlap in terms of the GO terms identified in this study and the Gordon et al. study. Indeed, many of the interactions reported by Gordon et al. may not be direct interactions due to the nature of the experimental protocol performed. One could easily imagine that a predicted interaction in this study may be involving other indirect interactions that were detected by Gordon et al. Hence, while the interactions do not overlap, the GO terms in which the proteins are involved in may overlap, since some of the proteins may be implicated in similar processes or machineries. Reporting such an overlap analysis would be interesting.

Additional comments

This is a very well written manuscript describing a comprehensive study that is of interest for the community performing research on SARS-CoV-2. The methods are sound, based on well-established tools, PIPE and SPRINT, and the work is also of high interest for the field of protein-protein interaction prediction in the context of poorly characterized organisms. The study provides information that could be used in the future for the development of therapeutics against SARS-CoV-2 and even other coronaviruses.

---

## Round 0.2 · accepted · Accept

Thank you for your attention to the reviewers comments and suggestions. Congratulations again!